

# Using SHAP to interpret XGBoost predictions of grassland degradation in Xilingol, China

Batunacun[1,2]*, Ralf Wieland[2], Tobia Lakes[1,3], Claas Nendel[2,3]

[1] Department of Geography, Humboldt-Universität zu Berlin, Unter den Linden 6, 10099 Berlin, Germany

[2] Leibniz Centre for Agricultural Landscape Research (ZALF), Eberswalder Straße 84, 15374, Müncheberg, Germany

[3] Integrative Research Institute on Transformations of Human-Environment Systems, Humboldt-Universität zu Berlin, Friedrichstraße 191, 10099 Berlin, Germany

* Correspondence to: Institute of Landscape Systems Analysis, Leibniz Centre for Agricultural Landscape Research (ZALF), Eberswalder Straße 84, 15374, Müncheberg, Germany

E-mail: batunacun@zalf.de



## Abstract

Machine learning (ML) and data-driven approaches are increasingly used in many research areas. XGBoost is a tree boosting method that has evolved into a state-of-the-art approach for many ML challenges. However, it has rarely been used in simulations of land use change so far. Xilingol, a typical region for research on serious grassland degradation and its drivers, was selected as a case study to test whether XGBoost can provide alternative insights that conventional land-use models are unable to generate. A set of twenty drivers was analysed using XGBoost, involving four alternative sampling strategies, and SHAP to interpret the results of the purely data-driven approach. The results indicated that, with three of the sampling strategies (over-balanced, balanced and imbalanced), XGBoost achieved similar and robust simulation results. SHAP values were useful for analysing the complex relationship between the different drivers of grassland degradation. Four drivers accounted for 99% of the grassland degradation dynamics in Xilingol. These four drivers were spatially allocated, and a risk map of further degradation was produced. The limitations of using XGBoost to predict future land-use change are discussed.

**Key words:** grassland degradation, machine learning, driver-driven method, XGBoost, SHAP values

## 1. Introduction

Land-use and land-cover change (LUCC) has received increasing attention in recent years (Aburas et al., 2019; Diouf & Lambin, 2001; Lambin et al., 2003; Verburg et al., 2002). Land-use change includes various land-use processes, such as urbanisation, land degradation, water body shrinkage, and surface mining, and has significant effects on ecosystem services and functions (Sohl & Benjamin, 2012). Grassland is the major land-use type on the Mongolian Plateau; its degradation was first witnessed in the 1960s. About 15% of the total grassland area was characterised as being degraded in the 1970s, which rose to 50% in the mid-1980s (Kwon et al., 2016). In general, grassland degradation (GD) refers to any biotic disturbance in which grass struggles to grow or can no longer exist due to physical stress (e.g. overgrazing, trampling) or changes in growing conditions (e.g. climate; Akiyama & Kawamura, 2007). In this study, grassland degradation is defined as grassland that has been destroyed and subsequently classified as some other land use, or that has significantly decreased in coverage.

Grassland is a land use that provides extensive ecosystem services (Bengtsson et al., 2019). When degraded, the consequences are seen in an immediate decline in these services, such as a decrease in carbon storage due to a reduction in vegetation productivity (Li et al., 2017). About 90% of carbon in grassland ecosystems is stored in the soil (Nkonya et al., 2016). Furthermore, GD results in a reduction in plant diversity and above-ground biomass available for grazing (Wang et al., 2014). Likewise, GD leads to soil erosion and frequent dusts storms in Inner Mongolia (Hoffmann et al., 2008; Reiche, 2014). Drivers of GD are manifold, and have been analysed in a range of studies (Li et al., 2012; Liu et al., 2019; Sun et al., 2017; Xie and Sha, 2012). However, few studies use sophisticated driver analysis to predict spatial patterns of GD (Jacquin et al., 2016; Wang et al., 2018). A number of studies have addressed the complex relationship between GD and its drivers (Cao et al., 2013a; Feng et al., 2011; Fu et al., 2018; Tiscornia et al., 2019a). However, these studies focus mainly on visualising or describing non-linear relationships between GD and its drivers.

The aim of developing various land-use models was to explore the causes and outcomes of land-use dynamics; these models were implemented in combination with scenario analysis to support land management and decision-making (National Research Council, 2014; Ren et al., 2019). Most such models are statistical models, such as logistic regression models or models based on principle





component analysis (Li et al., 2013; Lin et al., 2014) or Bayesian belief networks (Krüger and Lakes,
2015). Some such models are spatial (e.g. CLUE-S, GeoSOS-FLUS, LTM, Fu *et al.*, 2018; Liang *et*
*al.*, 2018; Pijanowski *et al.*, 2002, 2005; Verburg & Veldkamp, 2004; Zhang *et al.*, 2013); others
are not (e.g. Markov models; Iacono et al., 2015; Yuan et al., 2015). Hybrid models, which combine
different approaches to make the best use of the advantages of each model, are another important
variety. This type of model is used to characterise the multiple aspects of LUCC patterns and
processes (Li and Yeh, 2002; Sun and Müller, 2013). In most cases of land-use change, it was either
assumed that the relationship between the drivers and the resulting land-use change is constant over
time (Fu et al., 2018; Samie et al., 2017; Zhan J Y et al., 2007), or the relationships were identified
as being linear or non-linear, but were not interpreted (Tayyebi and Pijanowski, 2014a). We
hypothesise that the relationships between GD and its drivers are mainly non-linear. We therefore
see a need for methods that are capable of analysing and interpreting non-linear relationships
between GD and dynamic drivers.
With the development of computer science, machine learning (ML) models have been increasingly
used in land-use change modelling (Islam et al., 2018; Krüger and Lakes, 2015; Lakes et al., 2009;
Tayyebi and Pijanowski, 2014a). ML is superior to the human brain when it comes to pattern
recognition in large datasets, e.g. images and sensor fields. Once the task is defined and the data for
training is provided, ML operates without any further human assistance. Various ML approaches
have been used in the analysis of land-use change processes, the most prominent of which being
Support Vector Machines (SVM, Huang *et al.*, 2009, 2010), Artificial Neural Networks (ANN,
Ahmadlou *et al.*, 2016; Yang *et al.*, 2016), Classification And Regression Trees (Tayyebi and
Pijanowski, 2014b) and Random Forest (RF, Freeman *et al.*, 2016). While the different ML
approaches generally perform well in identifying patterns, they remain a black box and make no
contribution to our understanding of how the underlying drivers act on the LUCC process.
Compared to linear methods such as logistic regression, ML models often achieve higher accuracy
and capture non-linear land-use change processes. Likewise, ML models relax some of the rigorous
assumptions inherent in conventional models, but at the expense of an unknown contribution of
parameters to the outcomes (Lakes et al., 2009). However, the key challenge is to crack the black
box and reveal how each driver affects the land-use change pattern or processes in the ML models.
The eXtreme Gradient Boosting (XGBoost) method has recently been developed as a supervised
machine learning approach (Chen and Guestrin, 2016). XGBoost algorithms have achieved superior
results in many ML challenges; they are characterised by being ten times faster than popular existing
solutions, and the ability to handle sparse datasets and to process hundreds of millions of examples.
XGBoost has already been used in land-use change detection, combined with remote sensing data
(Georganos et al., 2018), but has not yet been used in the simulation and prediction of land-use
change. SHapley Additive exPlanations (SHAP; Lundberg & Lee, 2016) is a unified approach to
explain the output of any ML model and to visualise and describe the complex causal relationship
between driving forces and the prediction target. We propose using SHAP to analyse the driver
relationships hidden in the black box model of XGBoost when employed for land-use change
modelling.
Having earlier used a clustering approach to identify drivers of GD in a case study in Inner Mongolia
(Xilingol League; Batunacun *et al.*, 2019), we now use XGBoost and SHAP to simulate GD
dynamics across the same area. We are primarily interested in learning whether ML models can
achieve a better predictive quality than linear methods, in addition to improving our understanding
of how grassland degrades in Xilingol. In the intention to identify areas with a high risk of further
degradation and to determine the drivers responsible for progressive degradation, we used XGBoost
to generate a data-driven model to explore the GD patterns. We then used SHAP to open the non-
linear relationships of the black box model stepwise, and transformed these relationships into
interpretable rules. The resulting model enabled us to map the primary GD drivers and GD hot spots
in Xilingol.


## 2. Materials and Methods

### 2.1 Study area

The Xilingol League is located about 600 km north of Beijing (He et al., 2004), in the centre of Inner Mongolia. This administrative unit, covering an area of 206,000 km$^2$, spans from 41.4°N to 46.6°N and from 111.1°E to 119.7°E (Figure 1). The area is dominated by the continental temperate semiarid climate. The frequent droughts (in summer) and "dzud" (an extremely harsh and snow-rich winter) are the major natural disasters that occasionally lead to catastrophic livestock losses in this region (Allington et al., 2018; Tong et al., 2017; Xu GC et al., 2014). Xilingol possessed about 18,104 km$^2$ available pasture resources and 1240.4·10$^4$ sheep units at the end of 2015 (Xie and Sha, 2012). Around 1.044 million people lived in Xilingol in 2015, with ethnic Mongolian minorities accounting for around 31% and the rural population for 37% (Batunacun et al., 2019; Shao et al., 2017). Xilingol is a vast grassland, known for its high-quality meat products, nomadic culture, rich mineral resources and ethnic minorities. The ongoing degradation of grassland is receiving increasing attention. A set of economic stimuli and ecological protection policies launched in Xilingol were viewed as the root cause of GD over the past four decades. Although large-scale ecological restoration policies were implemented after 2000 in a bid to reduce GD, the problem still persists.

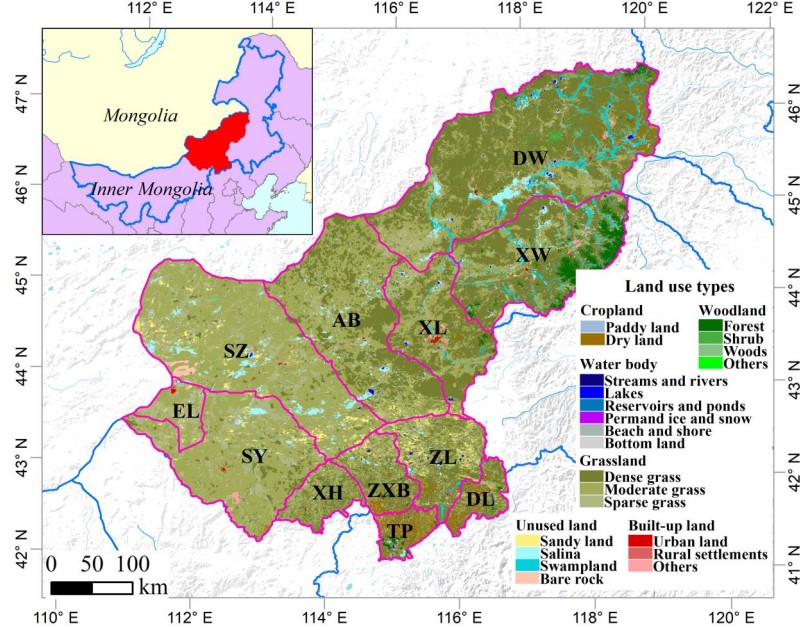

Figure 1: The location of the Xilingol League in Inner Mongolia and its land uses.

### 2.2 Grassland degradation

This study defines grassland degradation (GD) based on land-use conversion, involving two kinds of land-use change processes: (i) the complete destruction of grassland by transformation to another





type of land use (built-up land, cropland, woodland, water bodies and unused land), and (ii) a decline
in grassland coverage, which includes dense grass deteriorating into moderately dense grass and
sparse grass, and moderately dense grass deteriorating into sparse grass (see Fig. S 1a). Given that
GD is a dynamic process, we intended in this study to find the major drivers of newly added
grassland degradation (NGD). NGD refers to the difference in spatial GD extent between two
periods. About 13.0% of the total grassland area (176,410 km$^2$ in 2015) was degraded between 1975
and 2000 (Fig. S 1b); a further 10.6% was degraded in 2000-2015 (Fig. S 1c). Comparing the two
periods, approximately 10.2% of the grassland corresponded to the NGD area across the whole
region (Fig. S 1d). 18,093 pixels were extracted from the total NGD area, while the pixel number
of conversion for other land uses is 178,990 in this study (hereafter: non-NGD).

**2.3 Data collection**

In line with previous studies, a checklist of possible drivers (D) of GD was developed from the literature
(Cao et al., 2013b; Sun et al., 2017). A total of 19 drivers were grouped into four categories (see Table
1). All categories were described as follows: (1) Climate factors, including the annual mean temperature
(T) and annual sum of precipitation (P) in the growing season (April to Sep), were extracted from the
longest available weather dataset (from 1958-2015), in combination with evaluation data and the kriging
algorithm, to produce 1×1 km$^2$ raster files. (2) Geographic factors include elevation (DEM), and slope
and aspect (extracted from DEM data), which can be treated as the characteristic of each grid cell. The
DEM data were extracted from the SRTM 90m resolution and, after resampling, all data were processed
into 1×1 km$^2$ raster files. (3) Distance measures (the distance of each pixel centre to urban, rural, road
and mining, forest, cropland, dense grass, moderately dense grass, sparse grass and unused land pixels)
are widely used factors for different land-use models (Khoury, 2012; Samardžić-Petrović et al., 2016,
2017; Zhang et al., 2013). All distance measures were extracted from LUCC datasets from the years 2000
and 2015 using ArcGIS Euclidean distance, and processed into 1×1 km$^2$ grids. (4) Socio-economic
factors include the gross domestic product (GDP), sheep density and population density from 2000 and
2015. GDP and population density were obtained from a resources and environment data cloud platform,
CAS (http://www.resdc.cn/); sheep density data were accessed from statistical data; and we converted all
livestock data into grassland pixels. (5) Finally, we identified an area in which we assumed a strong
policy impact in the past, and developed a proxy for the policy effect on grassland degradation. Here, a
range of ecological protection measures were implemented inside and outside the Hunshandake and
Wuzhumuqin sand lands (see Fig. S 2), e.g. a livestock ban and the promotion of chicken farming (Su et
al., 2015). In a bid to explore policy effects, we assumed that GD is effectively slowed down by various
policies inside the sandy area (proxy set as 0), while outside the sandy area, land degradation is more
likely to happen in the absence of any policy effect (proxy set as 1, see Fig. S 2).





159    Table 1: Definition and derivation of drivers

| Code | Name of driver | Definition of driver | Unit | Measures | Time series | Original format | Process approach | Data sources |
|---|---|---|---|---|---|---|---|---|
| Climate factors | | | | | | | | |
| F1 | temperature | Difference between average temperature / total precipitation in growth season (April-September)in Phase 1* and Phase 2* | °C | Mean temperature | 2000, 2015-2030 | Grid | Kriging via ArcGIS and Python language | National Meteorological Information Center (https://data.cma.cn/) |
| F2 | precipitation | | mm | cumulative rainfall | 2000, 2015-2030 | | | |
| Geographic factors | | | | | | | | |
| F3 | DEM | DEM | m | -- | | Grid | -- | STRM |
| F4 | slope | slope | degree | -- | | Grid | Reclassification | http://srtm.csi.cgiar.org/SELECTION/inputCoord.asp |
| F5 | aspect | aspect | degree | -- | | Grid | Reclassification | |
| Distance measures | | | | | | | | |
| F6 | discrop | Change of distance to cropland in 2000 and 2015 | m | Distance | 2000, 2015 | SHP | Euclidean | Extraction from land-use data |
| F7 | disforest | Change of distance to forest in 2000 and 2015 | m | Distance | 2000, 2015 | | | |
| F8 | disunused | Change of distance to unused land 2000 and 2015 | m | Distance | 2000, 2015 | | | |
| F9 | disdense | Change of distance to dense grass 2000 and 2015 | m | Distance | 2000, 2015 | | | |
| F10 | dismode | Change of distance to moderate grass in 2000 and 2015 | m | Distance | 2000, 2015 | | | |
| F11 | dissparse | Change of distance to sparse grass 2000 and 2015 | m | Distance | | | | |
| F12 | disurban | Change of distance to urban in 2000 and 2015 | m | Distance | 2000, 2015 | | | |
| F13 | disrural | Change of distance to rural in 2000 and 2015 | m | Distance | 2000, 2015 | | | |





| F14 | disroad | Change of distance to road in 2000 and 2015 | m | Distance | 2000, 2015 | | | |
|---|---|---|---|---|---|---|---|---|
| F15 | dismine | Change of distance to mining in 2000 and 2015 | m | Distance | 2000, 2015 | | | |
| F16 | diswater | Change of distance to water in 2000 and 2015 | m | Distance | 2000, 2015 | | | |
| Social-economic factors | | | | | | | | |
| F17 | population density | Change of population density in 2000 and 2010 | Person | Person/ km2 | 2000, 2010 | Grid | Density | Resource and Environment data cloud platform, CAS. (http://www.resdc.cn/) |
| F18 | GDP* | Change of GDP in 2000 and 2010 | Yuan | Yuan/km2 | 2000, 2010 | Grid | Density | |
| F19 | sheep density | Change of sheep density in 2000 and 2015 | Sheep Unit | Sheep unit/km2 | 2000, 2015 | Grid | Density | Statistical data from Xilingol government website (http://tjj.xlgl.gov.cn/) |
| Scenario setting | | | | | | | | |
| F20 | policy | -- | -- | (0,1) | -- | Grid | -- | Assumption |

*Note: Phase 1 refers to 1975-2000; Phase 2 refers to 2000-2015. GDP: gross
domestic product.



### 2.3.1 XGBoost and logistic regression

Two algorithms were selected in this study: logistic regression (LR) and XGBoost. LR is a linear method involving two parts: the statistic LR and the classification LR. Both methods have already been used to simulate land use (Lin et al., 2011; Mustafa et al., 2018) and to define the relationship between land-use change and its drivers (Gollnow and Lakes, 2014; Mondal et al., 2014; Verburg et al., 2002; Verburg and Chen, 2000). Here, we use LR as a benchmark model to compare linear and non-linear methods in the simulation of land-use change. The optimised parameters of LG are C = 0.1, penalty = l2, solver = 'lbfgs', multi_class = 'multinomial'.

Boosting algorithms have been implemented in many past studies, where they often outperformed other ML algorithms (Ahmadlou et al., 2016; Filippi et al., 2014; Freeman et al., 2016; Keshtkar et al., 2017; Tayyebi and Pijanowski, 2014a). However, traditional boosting algorithms are often subject to overfitting (Georganos et al., 2018). To overcome this problem, Chen and Guestrin (2016) presented a new, regularised implementation of gradient boosting algorithms, which they called XGBoost (eXtreme Gradient Boosting). XGBoost was built as an enhanced version of the gradient boosting decision tree algorithm (GBDT), a regression and classification technique developed to predict results based on many weak prediction models – the decision tree (DT) (Abdullah et al., 2019; Freeman et al., 2016). XGBoost provides strong regularisation by adopting a stepwise shrinkage process instead of the traditional weighting process provided by GBDT. This process limits overfitting, minimises training losses and reduces classification errors while developing the final model (Abdullah et al., 2019; Hao Dong et al., 2018).

The XGBClassifier uses the following parameters: learning_rate (controls learning itself); max_depth (control depth of the RF); the n_estimators (controls the number of estimators used for the model); the min_child_weight (controls the complexity of a model, defines the minimum sum of weights of all observations required in a child); and lambda (L2 regularisation term on weights). The parameters were optimised using a simple grid search algorithm provided by scikit (Pedregosa et al., 2011) to estimate the optimal parameters (learning_rate = 0.1, max_depth = 9, n_estimater = 500, min_child_weight = 3, lambda = 10).

### 2.3.2 Sampling methods

Data are often distributed unevenly among different classes (Vluymans, 2019). Such imbalanced class distribution generally induces a bias. Canonical ML algorithms assume that data is roughly balanced in different classes. In real situations, however, the data is usually skewed, and smaller classes often carry more important information and knowledge than larger ones (Krawczyk, 2016). It is therefore important to develop learning from imbalanced data to build real-world models (Krawczyk, 2016; Vluymans, 2019). To ensure a highly accurate GD model, we introduced four different sampling methods in this study (Fig. S 3).

**Balanced sampling:** Random data sampling, resulting in equal sized samples.

**Imbalanced sampling**: Random data sampling, but with the same share of the sampled class, resulting in unequal sized samples.

**Over-sampling:** Artificial points are added to the minority class of an imbalanced sampling set, making it equal to the majority class and resulting in equal sized samples.

**Under-sampling:** Points are removed from a majority class of an imbalanced sampling set, making it equal to the minority class and resulting in equal sized samples (He and Garcia, 2009).




In the present study, we used these four sampling methods to evaluate the model in the context of
the sampling method and its performance in the training process and the simulation process (see Fig.
S 3). In our case study, 18,190 pixels (about 10% of the total) were selected by different sampling
methods (Fig. S 3) to train (66% of the sample size) and test (34% of the sample size) the model.

### 2.3.3 SHAP values

SHAP (SHapley Additive exPlanations) is a novel approach to improve our understanding of the
complexity of predictive model results and to explore relationships between individual variables for
the predicted case (Lundberg and Lee, 2017). SHAP is a useful method to sort the driver's effects,
and break down the prediction into individual feature impacts. Feature selection is of primary
concern when using ML methods to process land-use change (Samardžić-Petrović et al., 2015, 2016,
2017). SHAP values show the extent to which a given feature has changed the prediction, and allows
the model builder to decompose any prediction into the sum of the effects of each feature value and
explain – in our case – the predicted NGD probability for each pixel (see Figure 3). In this study,
we used SHAP values to sort the driver's attributions; capture the relationship between drivers and
NGD; and map the primary driver for NGD at the pixel level.

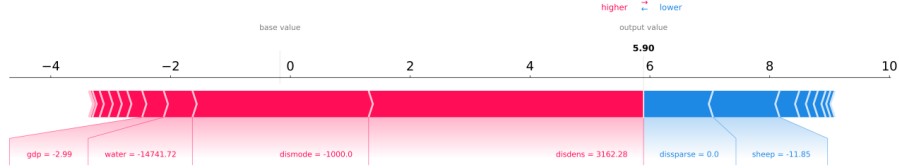


Figure 2: Decomposed SHAP values for the individual prediction of an example pixel.
In our study, we define the *base value* as the value that would be predicted by the model if no feature
knowledge were provided for the current output (mean prediction); we define the output value as
the prediction for this particular observation. SHAP values are calculated in log odds. Features that
increase the value of the prediction (to the left in Fig. 2) are always shown in red; those that lower
the prediction value are shown in blue (to the right in Fig. 2, Dataman, 2019). In this instance (Figure
2), *disdense* (change of distance to dense grass) is the primary driver of NGD at this pixel level
(largest value). The fact that the value is positive means that the risk of NDG increases in line with
an increase in distance to dense grass areas.

### 2.3.4 Validation of the model

Two validation steps are required for ML models: validation of the training process, and validation
of the simulation process. For the training process, a robust model was selected using overall
classification accuracy, precision, recall and the kappa index. Accuracy, precision and recall were
calculated based on a confusion matrix (CM) (He and Garcia, 2009). For the simulation process, the
final model was validated using the kappa index, the area under the precision-recall curve, and recall.
The validation indicators are defined as follows.
Overall classification accuracy (ACC) is the correct prediction of NGD and other pixels in the whole
region. This indicator was used to evaluate the accuracy of the model. Precision is the proportion of
correctly predicted positive examples (refers to NGD in this study) in all predicted positive examples.
Recall is the proportion of correctly predicted positive examples in all observed positive examples
(the observed NGD) (Sokolova and Lapalme, 2009). In general, high precision predictions have a
low recall, and vice versa, depending on the predicted goals. Here, since we focus on NGD and
other land-use changes, we use both indicators to evaluate our models.





Table 2: Confusion matrix for binary classification of newly added grassland degradation (NGD) and
other changes, including four indicators: false positives (FP), cells that were predicted as non-change but
changed in the observed map; false negatives (FN), cells that were predicted as change, but did not
change in the observed map for disagreement; true positives (TP), cells that were predicted as change
and changed in the observed map; and true negatives (TN), cells that were predicted as non-change and
did not change in the observed map for agreement.

| | | Observed values | | |
|---|---|---|---|---|
| | | Others | NGD | |
| Simulated values | Others | True negatives (TN) | False positives (FP) | Recall=TP/ (TP+FN) |
| | NGD | False negatives (FN) | True positives (TP) | |
| | | Precision =TP/(TP+FP) | | |
| | | ACC=(TP+TN)/(TP+FN+FP+TN) | | |

The precision-recall curve (PR curve) provides more information about the model's performance
than, for instance, the Receiver Operator Characteristic curve (ROC curve), when applied to skewed
data (Davis and Goadrich, 2006). The PR curve shows the trade-off of precision and recall, and
provides a model-wide evaluation. The area under the PR curve (AUC-PR) is likewise effective in
the classification of model comparisons. The baseline for the PR curve (y) is determined by positives
(P) and negatives (N). In our study, y = 0.09 (y = 18374/200652), which means when AUC-PR =
0.09, the model is a random model (Brownlee, 2018; Davis and Goadrich, 2006).
The kappa index (κ) is a popular indicator used to measure the proportion of agreement between
observed and simulated data, especially to measure the degree of spatial matching. When κ > 0.8,
strong agreement is yielded between the simulation and the observed map; $0.6 < κ < 0.8$ describes
high agreement; $0.4 < κ < 0.6$ describes moderate agreement; and $κ < 0.4$ represents poor agreement
(Landis and Koch, 1977).
In this study, κ was used to evaluate the agreement and disagreement between observed NGD and
simulated NGD. Kappa should be the primary validation measure, followed by AUC-PR (used to
evaluate model performance) and recall (used to evaluate model sensitivity). Features and
definitions of these indicators are given below.
**2.3.5 The structure of the ML model**
The ML methodology of simulating GD involves six steps (Fig. S 4): (1) Target definition and data
collection and processing; the targets of this study are to build a robust ML model for simulating
NGD, as well as visualising these complex relationships between various variables and the dynamics
of GD. A total of 20 drivers (D) of GD were collected. All dynamic drivers were processed by GIS
into raster files and exported into ASCII files as final inputs for the ML model. (2) Data organisation:
the ML model simulates land-use change as a classification task (Samardžić-Petrović et al., 2015,
2017). In the present study, we organise this task as a binary classification $Y$ ( value 1 and 0, stand
for NGD and Non-NGD); related drivers are $x$ ($x_1, x_2, x_3 \ldots \ldots x_n$), $n$ is the driver identifier, and $x$
denotes the change in value of each driver. The process of data standardisation is usually necessary
for most ML models, but since XGBoost is a tree-based method, it does not require standardisation
or normalisation. In this case, we performed standardisation only for the logistic regression model.
(3) Data sampling: this is a necessary step to avoid overfitting or the loss of important information.
The sampling method generally includes balanced and imbalanced sample strategies. In this study,
we tested various balanced sampling strategies to identify the most suitable one. (4) Model building
and selection: a ranking was used to find the best model in each specific case. In our study, we
defined a model with κ > 0.8 and AUC-PR>0.09 as robust, while $0.6<κ<0.8$ and AUC-PR>0.09
represents an acceptable model. (5) Model validation and feature ranking: after tuning the model,
the most robust model and the driver with most useful information are selected. (6). The last step is




explaining the model and the simulation.

# 3. Results

**3.1 Model validation**

The XGBoost model outperformed the LG model in both training and simulation (Figure 3 and 4).
The LG model seems to be an inappropriate model for understanding NGD in this case. XGBoost
yielded robust results in both training and simulation, with indicator values almost entirely above
90%.

Figure 3 indicates that XGBoost performed very well across all balanced sampling methods (over-
sampling, under-sampling and balanced sampling, red rectangle in Figure 3) in the training process.
Only the imbalanced sampling exhibited a slightly weaker performance in the training process. This
is mainly due to the balanced sampling datasets, which provided more information for the model.
In addition, the model was affected less than the imbalanced sampling method by the majority class
or unchanged cells (Mileva Samardzic-Petrovic et al., 2018).

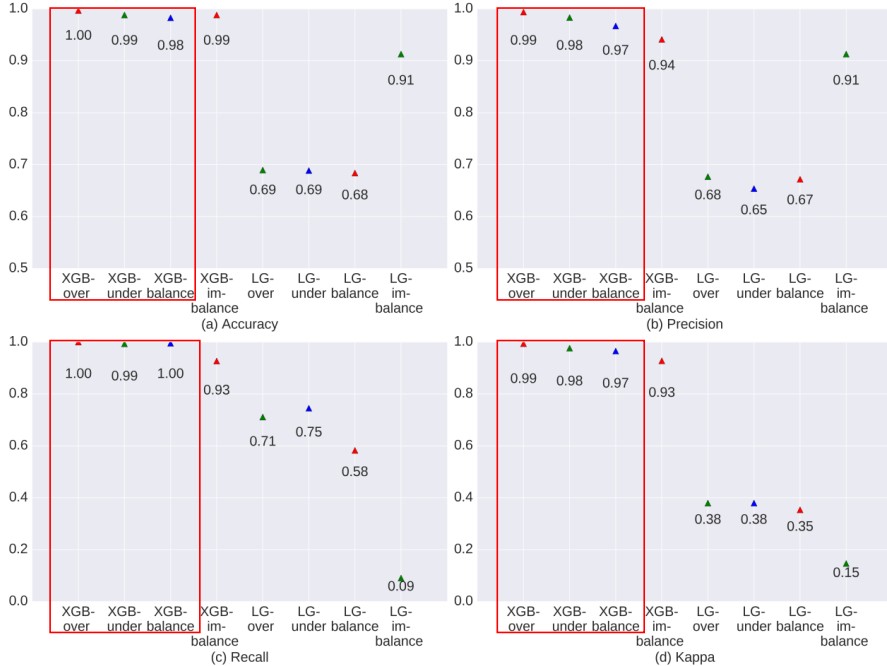

Figure 3: Evaluation of model performance during the training process.
Figure 4 and Figure 5 show the model evaluation results in the simulation process and the spatial
prediction maps. XGBoost with under-sampling (green rectangle in Figure 4) yielded the weakest
performance compared to the other three sampling methods. This is mainly due to the smaller
sample size, which prevents the model from extracting sufficient experience. As can be seen in
Figure 5b, XGBoost used with the under-sampling method produced the error map with the highest





FP values, where the model predicted non-change points as change points. The under-sampling
method is unable to identify NGD points sufficiently well. XGBoost used with the over-sampling
method caused balanced and imbalanced sampling to have similar and strong prediction abilities
(see Figure 4), differing only slightly in their CM indicators (see Figure 5). We finally selected
XGBoost combined with the over-sampling strategy for our study, mainly because of its relatively
higher values in κ, AUC-PR and recall (see Figure 4).

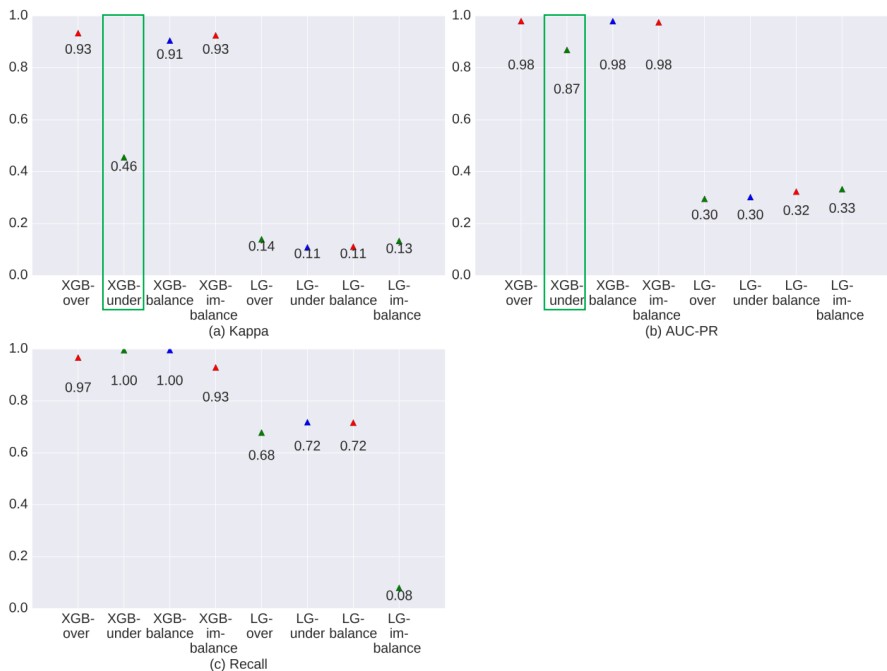

Figure 4: Evaluation of model performance during the prediction process.

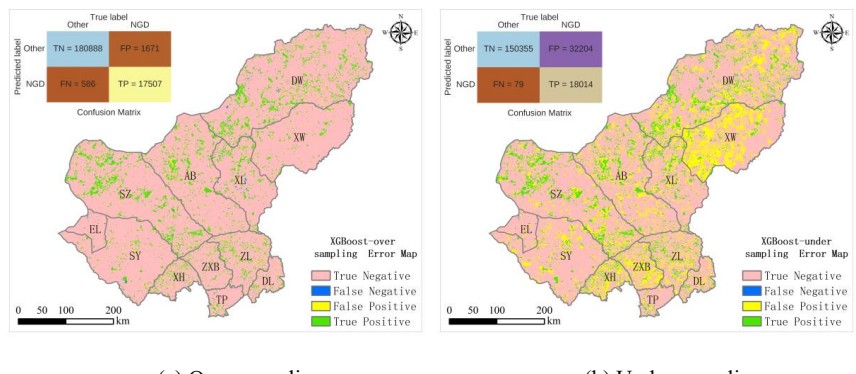

(a) Over-sampling                    (b) Under-sampling

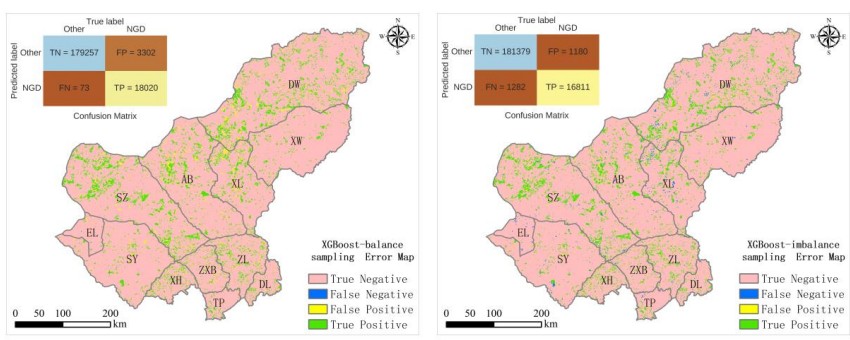

(c) Balanced sampling                    (d) Imbalanced sampling

Figure 5: Error map of different sampling methods using the XGBoost model.

**3.2 Driver selection**

Figure 6 is a summary plot produced from the training dataset; it includes approximately 13,200 points (66% of the sample size). This plot combines feature importance (drivers are ordered along the y-axis) and driver effects (SHAP values on the x-axis), which describe the probability of NGD having occurred. Positive SHAP values refer to a higher probability of NGD. The gradient colour represents the feature value from high (red) to low (blue), as previously introduced in Figure 2. As Figure 6 shows, *disdense* was the primary driver for NGD in the study region. The relationship between *disdense* and NGD is non-linear, which can be seen from the SHAP values being both positive and negative (black rectangle in Figure 6). The interpretation of the effects of *disdense* can be summarised as a higher probability of NGD with increasing distance from dense grassland (see black rectangle in Figure 6 with pink colour on the right).

Figure 6 shows that driver effects include both linear-dominated relationships, such as *sheep*, *GDP* and others, and non-linear-dominated relations, such as *disdense*, *dismode* and others. In addition, the figure shows that the most important drivers for NGD are the changes of distance to dense, moderately dense and sparse grassland, then followed by sheep density and the distance to unused land. The effect of policies comes almost at the bottom, indicating that policies implemented outside sandy areas seem to have little effect on GD. The geographical factors DEM and slope are also positioned mid-field. The effect of geographical drivers does not appear to be as strong as the effect of other drivers. The change of distance to mining, located at the bottom for all drivers, does not have a strong effect on NGD compared to other drivers.

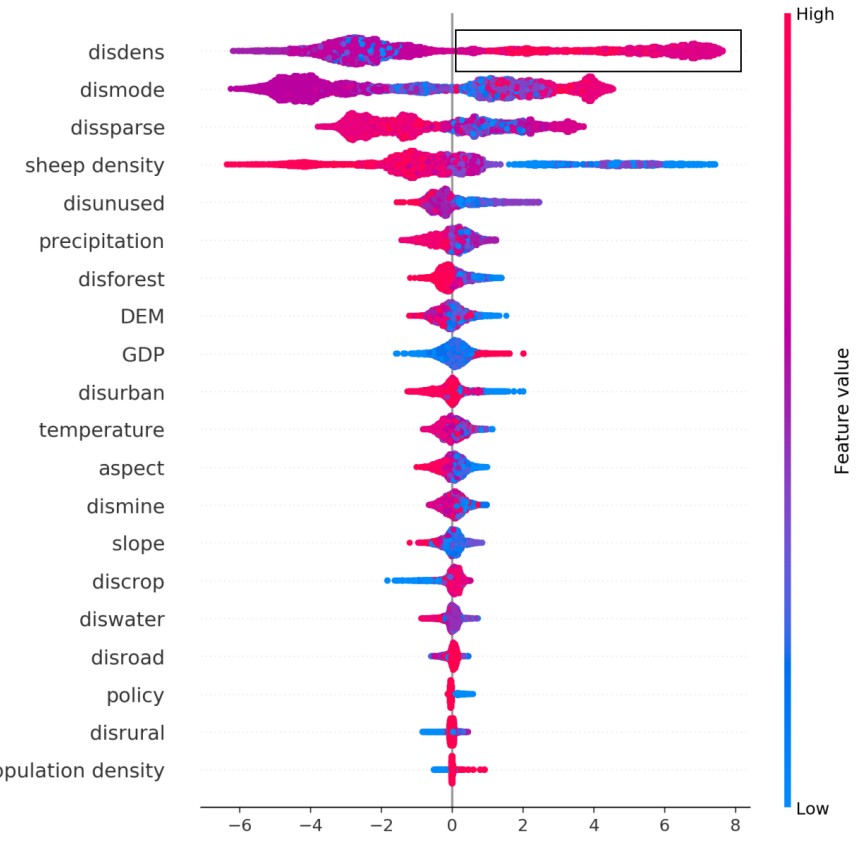

Figure 6: Driver ranking by SHAP values based on the training dataset (66% of sample size) using the
over-sampling method.

Note: The top rank indicates the most significant effects across all predictions. Each point in the cloud to
the left represents a row from the original dataset. The colour code denotes high (red) to low (blue) feature
values. Positive SHAP values represent a higher likelihood of NGD, while negative values indicate lower
likelihoods. The range across the SHAP value space indicates the degradation probability, expressed as
the logarithm of the odds.

A recursive attribute elimination method was performed to determine how attribute reduction affects
modelling performance using XGBoost with the oversampling method (see Fig. S 5; for more details,
refer to Samardžić *et al.*, 2015). The results indicate that the first three drivers may already produce
a satisfactory model ($\kappa$ = 0.74, AUC-PR = 0.85, recall = 0.92), while adding the fourth driver can
produce a robust model ($\kappa$ = 0.94, AUC-PR = 0.98, recall = 0.98). This means that XGBoost used
with the oversampling strategy can predict NGD with very high accuracy using a relatively small
amount of data. Fig. S 6 shows the simulation result using the first four drivers, and compares the
results with the observed map.

**3.3 Relationship between NGD and drivers in the XGBoost model**

SHAP values and spread (Figure 7) indicate that no linear relationship between driver and prediction





could be found for any of the individual features. Change of distance to dense, moderately dense
and sparse grass pixels, and change of sheep density were the dominant drivers for NGD. Figure 7a
indicates that when *disdense* < 0, the SHAP value is negative, and when the distance to dense grass
areas is small, the likelihood of degradation is also small. The relationship seems to be more
complex for distance to moderately dense grass (*dismode*, Figure 7b); here, no simple linear
interpretation is obvious. For distance to sparse grass (*dissparse*, Figure 7c), the pattern again
suggests a rather linear interpretation, which is that the likelihood of degradation increases with
decreasing distance. For sheep density, Figure 7d indicates that when sheep density decreased, the
probability of GD obviously increased. Policy was not identified as a major driver of GD (Figure
6). However, policy effects obviously have a different impact inside and outside sandy zones. Figure
7e shows that our initial assumption is invalid: the probability of GD increased inside the sandy
areas where we assumed effective policy measures to be in place (value 0). This result is also in line
with Figure 7g, which shows that the closer to unused land, the more likely degradation will occur.
We can identify three groups for the remaining 14 drivers. For GDP and population density (Figure
7g and Figure 7h), the likelihood of NGD increases with increasing values. Figure 7i-j indicate that
warmer and drier climate conditions increase the probability of GD. Figure 7k, l, m and n indicate
that the probability of GD rises with closer distances to forest, urban, rural and water areas. Figure
7o shows a slight SHAP value pattern, in which the closer to cropland, the more unlikely degradation
will occur. This is mainly due to transformation from cropland to grassland. Figure 7p-t do not show
any interpretable spatial pattern.

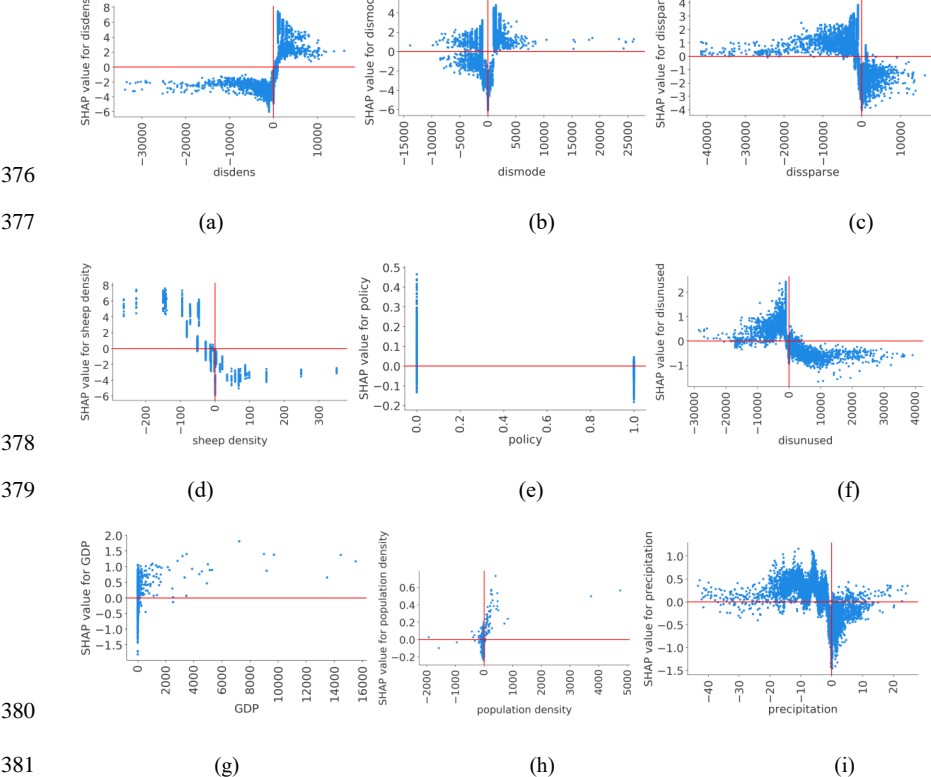


(a)                              (b)                              (c)

(d)                              (e)                              (f)


(g)                              (h)                              (i)




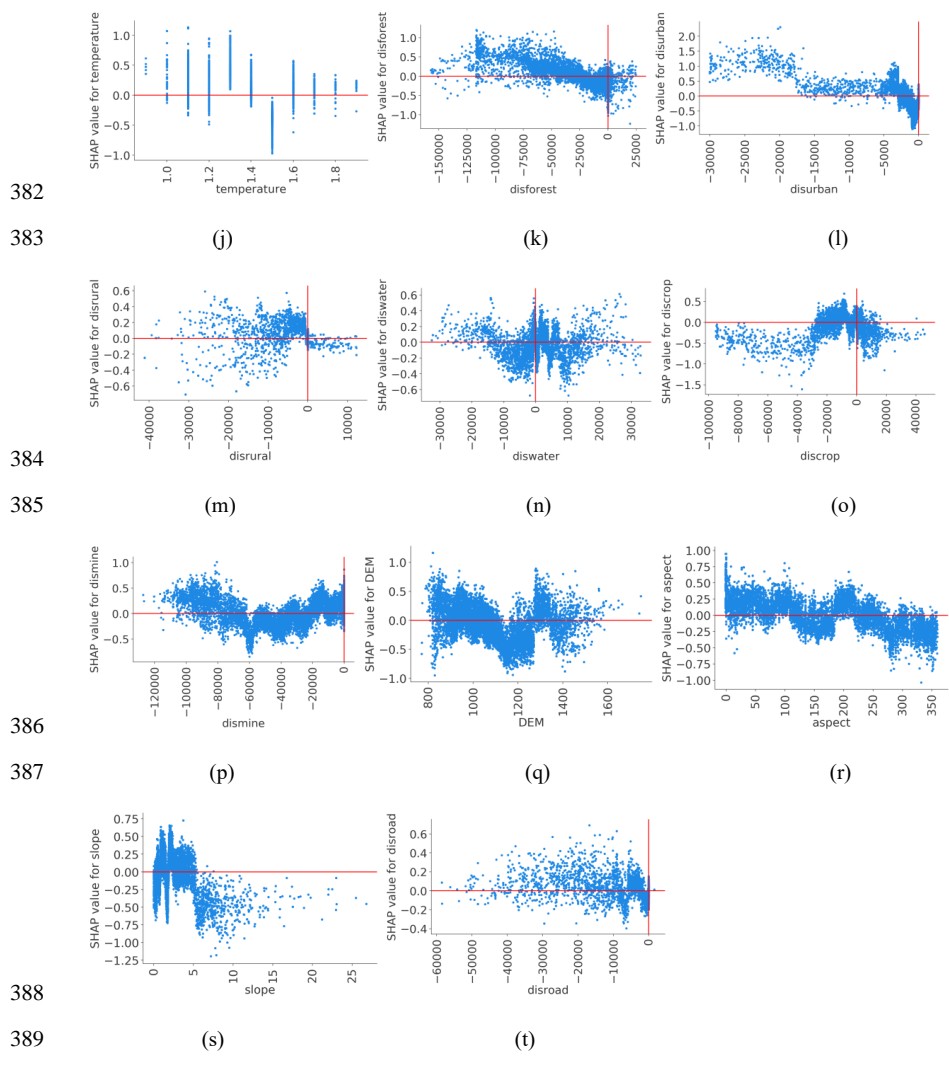


(j)                 (k)                 (l)


(m)                (n)                 (o)


(p)                (q)                 (r)


(s)                (t)

Figure 7: The SHAP dependence plot for each driver.

**3.4 Mapping the primary drivers of NGD**
All drivers' contributions to NGD were ranked according to their SHAP values for each pixel in this
study. Figure 8 shows the primary driver for each NGD pixel. Distance to grassland pixels (dense,
moderately dense and sparse grass) were the major drivers of NGD, responsible for 9,478, 3,892
and 1,629 NGD pixels, respectively. Sheep density was responsible for 3,042 NGD pixels, ranking
third among all drivers. This order differs to that in Figure 6 and Figure 8 because in those cases,
ranking is based on the total contribution of all drivers. Fig. S 7 shows the number of NGD pixels
in which a driver was dominant or primary. The change of distance to any type of grassland was the
primary driver for about 82.8% of the total NGD pixels; sheep density accounted for 16.8%. The





remaining seven drivers caused less than 1% of the total NGD. We can see from the spatial pattern
that the change of distance to grassland was the major driver for GD in the dense grassland region
(counties of DW, XL and AB), while in the counties of SZ, SY, ZXB, ZL and TP, sheep density was
often identified as the major driver.

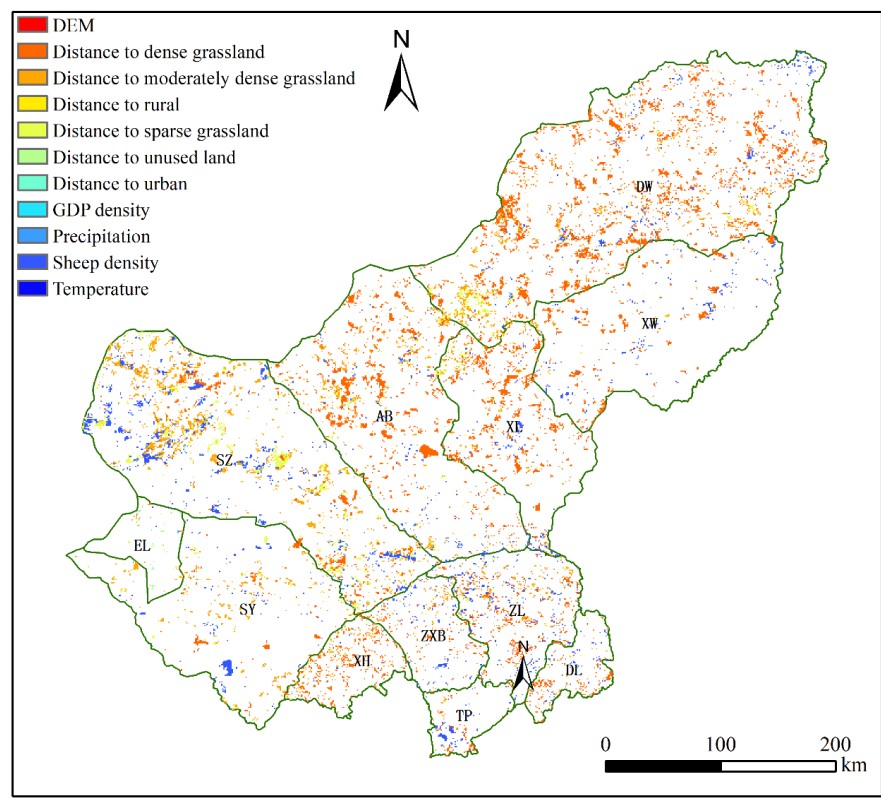


Figure 8: Spatial patterns of primary drivers for each pixel.

**3.5 Regions of high risk for grassland degradation**

A probability map of NGD was produced (Figure 9). Low probabilities of NGD were found in the
central and northern counties (DW, XL, AB, SZ, ZL ZXB and XH), while high probability regions
were EL, SY and XW. TP and DL in the south were categorised as low probability regions, due to
their lower share of grassland area.

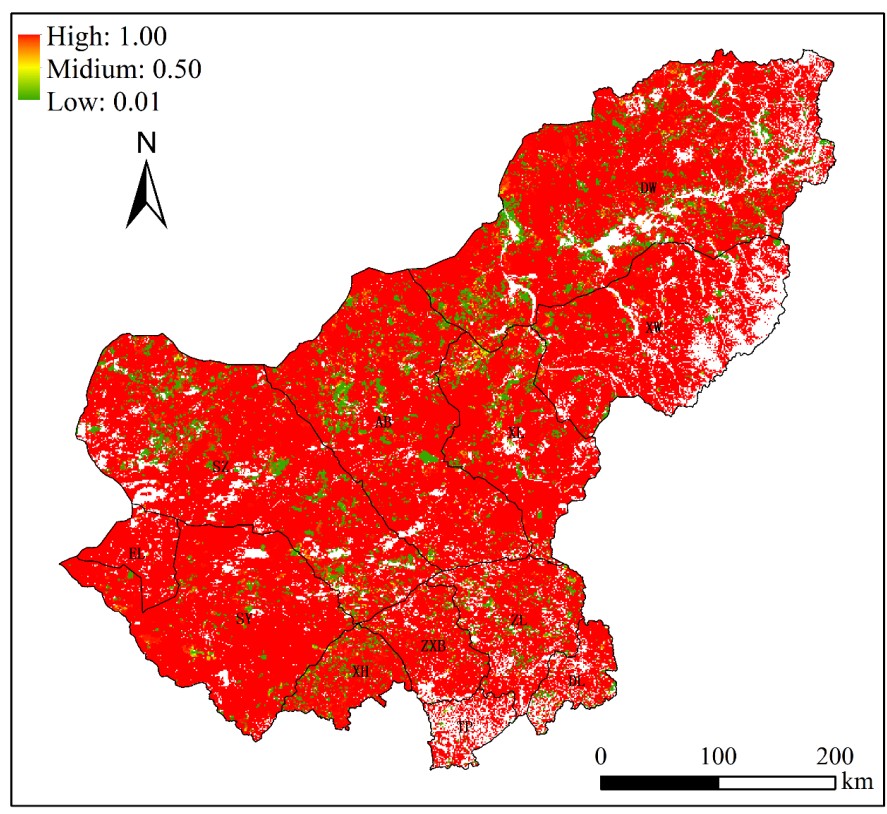


Figure 9: Degradation probability map for grassland in Xilingol.
## 4. Discussion
### 4.1 ML model building and evaluation
In this study, we defined a general framework for creating an ML model using the XGBoost
algorithm for the purpose of analysing and predicting land-use change. XGBoost obtained a κ of
93% and a recall value of > 99% when used to simulate and predict GD in this study. Compared to
other popular ML learning algorithms, XGBoost exhibited a strong prediction ability. In studies
where ANN, SVM, RF, CART, Multivariate Adaptive Regression Spline (MARS) or LR were used
in combination with Cellular Automata (CA), the recall value is usually 54%-60% (Shafizadeh-
Moghadam et al., 2017). Ahmadlou *et al.* (2019) stated that MARS and RF only yield high accuracy
in training runs, but do not prove very accurate in the validating process when simulating land-use
change.
Concerning the four sampling strategies we used to test the imbalance issue, we found that all
strategies performed satisfactorily in the training runs. In the simulation, the under-sampling
strategy yielded a relatively low accuracy (κ = 0.46) model. We assume that removal of data from
the majority class causes the model to lose the important concepts pertaining to the majority class





(He and Garcia, 2009). XGBoost used with the under-sampling method always produced similar
results, irrespective of the size of the dataset (see Fig. S 8). We conclude from this pattern that
XGBoost is also able to use sparse data to reflect real-world problems (Chen and Guestrin, 2016).
**4.2 SHAP values and drivers of grassland degradation**
The general idea of introducing SHAP values as a further tool to analyse XGBoost ranking is to
provide a method to evaluate the ranking with respect to causal relationships. The original XGBoost
ranking is based on the in-built feature selection functions *Gain* (refers to the improvement in
accuracy provided by a feature), *Weight* (or frequency, refers to the relative number of a feature
occurrence in the trees of a model) and *Coverage* (refers to the relative numbers of observations
related to this feature). However, these functions always produce different rankings of drivers (Abu-
Rmileh, 2019) due to random components in the algorithms. SHAP values introduce two further
properties of feature importance measures: *consistency* (whenever we change a model such that it
relies more on a feature, the attributed importance for that feature should not decrease) and *accuracy*
(the sum of all feature importance values should equate to the total importance of the model;
Lundberg, 2018; Lundberg & Lee, 2017). Consistency is required to stabilise the ranking throughout
the analysis, reducing the change of order in the ranking to a minimum when the number of
identified drivers changes. The accuracy property of SHAP makes sure that each driver's
contribution to overall accuracy remains the same, even when drivers are excluded from analysis.
Other methods usually compensate for the withdrawal of a driver from the analysis, which makes
the determination of a single driver's contribution difficult.
The feature ranking based on SHAP values indicated that the change of distance to any type of
grassland (dense, moderately dense and sparse grass) is the most important driver for any newly
added grassland degradation. In this context, dense and moderately dense grassland areas are more
easily degraded than other land-use types, followed by sparse grass. These results are in line with
previous studies (Li et al., 2012; Xie and Sha, 2012). Good-quality grassland is more likely to be
degraded through increasing human disturbance. An explanation for this can be derived from local
people's living strategies. People who live in good-quality grassland areas are more likely to use
grassland for livestock production with higher animal densities, risking overgrazing. Furthermore,
Li *et al.* (2012) indicated that good-quality grassland is more likely to be converted to other land-
use types, such as cropland. In contrast, people who have lived in sparse grassland regions for
centuries have long adapted to low productivity, reducing their livestock numbers accordingly. They
have also developed strategies to cope with variability in weather conditions, e.g. by preparing and
storing more fodder and forage.
Sheep density was identified as the fourth major driver. However, the SHAP values indicate that
when sheep density decreases, the probability of grassland degradation increases. Overgrazing has
been the dominant driver for grassland degradation on the Mongolian plateau before, which has
changed the grassland ecosystem significantly towards lower grass coverage (Nkonya et al., 2016;
Wang et al., 2017). However, there is recent evidence that this causal relationship has changed. It
now appears that farmers increasingly select their livestock numbers according to the carrying
capacity of the grazing land (Cao et al., 2013b; Tiscornia et al., 2019b). By passing the "Fencing
Grassland and Moving Users" policy (FGMU), the Chinese government issued a law that regulates
livestock numbers based on a previously calculated carrying capacity. This development has
upturned the causal relationship between livestock numbers and NGD, reflected by the SHAP value
pattern in Figure 6.
Besides the four main drivers, seven other drivers also occasionally appear as the main driver for
some pixels (Figure 8). This highlights the fact that, at the local level, other drivers apart from the
four drivers identified as being major can also play a significant role. For example, in the county of
EL, the remaining seven drivers were mainly responsible for NGD. EL has less NGD after 2000





compared with other counties in Xilingol (Fig. S 1), and most of the EL area is covered by sparse
grass. EL is the most frequented border control point to Mongolia, and is subject to intensive tourism.
In the sparse grassland and agro-pastoral regions (SZ, SY, ZXB, ZL and TP), sheep density was
identified as the important driver. This indicates that, even though livestock numbers have decreased,
grassland is still experiencing serious degradation in this region. Here we see additional potential
for installing further grassland conservation measures, such as adjusting the livestock number to the
grassland carrying capacity.

**4.3 The current risk of grassland degradation in Xilingol**

Three regions of different risk classes were identified in the probability map of NGD (Fig. 9). The
low-risk region (DW, XL, AB, SZ, ZL ZXB and XH) is dominated by good-quality grassland (dense
and moderately dense grass). In recent decades, this region has suffered from increasing human
disturbance, e.g. overgrazing and mining development. However, after 2000, grassland in this region
has recovered, mainly as the result of ecological protection projects (Sun et al., 2017). Even though
this region is predicted as being less exposed to the risk of land degradation in the future, attention
is still required for the restoration process. The high-risk region includes the counties of EL, SY and
XW. EL and SY are covered by a large share of low-quality grassland, which – due to its own
fragility – is likely to be affected by extreme climate and human disturbance, more than, e.g. higher-
quality grasslands. The recent change in grassland property rights and the establishment of
ecological protection projects have also limited the mobility of nomadic herders throughout Xilingol.
As a consequence, herders cannot easily change grazing spots if extreme weather occurs; they are
then bound to have their cattle graze at the same spots, increasing the pressure on low-quality
grasslands in particular (Qian, 2011). For a long time, fragile grassland remained in an equilibrium
state with the extreme weather (frequent droughts, "dudz") to which it was exposed, and with the
nomadic livestock husbandry that the region's inhabitants practised. However, when the property
rights of grassland and livestock were changed from collective to private, the nomadic lifestyle was
largely abandoned.

**4.4 The limitations of XGBoost for scenario exploration**

XGBoost has already scored top in a range of algorithm competitions in the data scientists
community (Kaggle, 2019) due to its high accuracy and speed (Chen and Guestrin, 2016). ML
models extract patterns from data, without considering any existing expert knowledge, which is why
they are increasingly used to identify non-linear relationships (Ahmadlou et al., 2016; Samardžić-
Petrović et al., 2015; Tayyebi and Pijanowski, 2014b). However, ML models require specific data
structures for each problem to which they are applied. In this study, we simulated grassland
degradation in two different phases (1975-2000 and 2000-2015). All time series of driver data were
organised as model inputs, while grassland degradation dynamics were organised as prediction
targets. Although the model achieved high accuracy in predicting NGD in Phase 2, it was not
possible to achieve acceptable results in simulating both Phase 1 and Phase 2 separately. Second,
compared with conventional models, the XGBoost model cannot be easily transferred to other
regions for the same research question. Models like CLUE-S and GeoSOS-FLUS have been widely
used in different regions across the world (Fuchs et al., 2017; Liang et al., 2018a; Liu et al., 2017;
Verburg et al., 2002). When ML models are used in other regions, driver data must be collected and
structures adapted. Thirdly, ML models always need to learn sufficiently before they are able to
make predictions. This requires a sufficient amount of data covering historical periods or different
land-use change patterns.
XGBoost alone is unable to project any scenarios of land-use change based on historical data.
However, the methodology presented here can be applied to quantify alternative scenarios produced
using other approaches, such as conventional, rule-based models (Verburg et al., 2002) or cellular


automata (Islam et al., 2018; Shafizadeh-Moghadam et al., 2017).

## 5 Conclusion

Machine learning and data-driven approaches are becoming more and more important in many research areas. The design and development of a practical land-use model requires both accuracy and predictability to predict future land-use change, a well-fitted model that reflects and monitors the real world (Ahmadlou et al., 2019). The method framework presented here for building an ML model and explaining the relationship between drivers and grassland degradation identified XGBoost as a robust data-driven model for this purpose. XGBoost showed higher accuracy in training and simulation compared to existing ML models. Combined with over-sampling, it slightly outperformed in the simulation process. The simulated map has a high agreement with the observed values (kappa=93%).

We identified six basic steps that should be included in ML model building, and they are also similar for other research applications (Kiyohara et al., 2018, 2018; Kontokosta and Tull, 2017; Subramaniyan et al., 2018). However, different validation measures can be introduced in both the training process and the simulation process. In this study, we tested different evaluation measures to evaluate the ML model, e.g. a typical confusion matrix to evaluate the training process, AUC-PR to evaluate the goodness of the ML model, and the kappa index to measure the degree of matching between observed and simulated values. These validation indicators consider both the research object and data characteristics. For example, when the data size is unbalanced, AUC-PR is a better choice than AUC-ROC (Brownlee, 2018; Davis and Goadrich, 2006; Saito and Rehmsmeier, 2015).

SHAP was introduced in this context to provide a causal explanation of the patterns identified by the ML model. In our case, SHAP was used to explain how drivers contribute to grassland degradation processes at the pixel and regional level, despite their non-linear relationship. According to the analysis, the distance to dense, moderately dense, and sparse grass, and sheep density, were the most important drivers that caused new grassland degradation in this region. In addition, individual SHAP values of sheep density indicated that the causal relationship between grassland degradation and livestock pressure has changed over time: the increase in sheep density was not the major driver for NGD in Phase 2 of the land degradation trajectory. Instead, the decrease in the grazing capacity of grassland caused a decrease in livestock numbers. The primary driver map of NGD provided a more detailed picture of NGD drivers for each pixel, as an important support for grassland management in the Xilingol region. The individual SHAP values of each driver may be an important prerequisite for rule-based scenario-building in the future.

**Author contribution:**

Batunacun prepared the manuscript with contribution from all co-authors, and Batunacun performed the simulation. Ralf Wieland develop the model code.

**Code and data availability**

The development of XGBoost and SHAP values, graphs and model validation presented in this paper were conducted in Python language. The original source code for XGBoost was obtained from website (https://xgboost.readthedocs.io/en/latest/) and the explanation of XGBoost could be obtained from Chen et al (2016). The original source code of SHAP values could be found in GitHub (https://github.com/slundberg/shap), while the explanation could be obtained from Lundberg et al (2017). The land use data in this manuscript can be downloaded Resource and Environment data cloud platform, CAS (http://www.resdc.cn/), the statistical data could be obtained form Inner



Mongolia statistic year book. Other data get from various data sources, please check Table 1.

**Competing interests:**
The authors declare that they have no conflict of interest
**Acknowledgements**
The authors express their sincere thanks to the China Scholarship Council (CSC) for funding this
research and to Elen Schofield for language editing.

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
