# Peer review of "Using SHAP to interpret XGBoost predictions of grassland degradation in Xilingol, China"

_Geoscientific Model Development, 2020_

## Short Comment (SC1) · 22 Jun 2020

Dear authors,

in my role as Executive editor of GMD, I would like to bring to your attention our Editorial version 1.2:

https://www.geosci-model-dev.net/12/2215/2019/

This highlights some requirements of papers published in GMD, which is also available on the GMD website in the 'Manuscript Types' section:

http://www.geoscientific-model-development.net/submission/manuscript_types.html

Therefore,

- as the python scripts are essential for you work, these python scripts need to be archived persistently(DOI) and made public available.

- as XGBoost and SHAP are sufficiently specialist that the model wouldn't make sense without them, they also need to be archived (a github link does not cut it).

- You haven't identified any of the input data with sufficient specificity that the work could be reproduced. That needs to change. Mostly you just provide top level organisation websites as data sources. You need to provide proper data citations.

Yours,

Astrid Kerkweg

---

## Referee Comment (RC1) · Anonymous Referee #1 · 14 Jul 2020

This manuscript tests whether XGBoost can provide alternative insights that conventional land-use models are unable to generate. The overall methodology is interesting. I have a number of major comments before I can suggest the paper for publication.

-Line 54: "Some such models are spatial (e.g. CLUE-S, GeoSOS-FLUS, LTM, Fu et al., 2018; Liang et 55 al., 2018; Pijanowski et al., 2002, 2005; Verburg & Veldkamp, 2004; Zhang et al., 2013); others are not (e.g. Markov models; Iacono et al., 2015; Yuan et al., 2015)." Authors should be aware that all land use change models are spatial models. Markov models are used to estimate the quantity of change from one land use state to another but are not land use change simulators.

-Line 57: "Hybrid models, which combine different approaches to make the best use of the advantages of each model, are another important variety. This type of model

is used to characterise the multiple aspects of LUCC patterns and processes (Li and Yeh, 2002; Sun and Müller, 2013)." Authors did not discuss important other land use modeling approaches such as Cellular Automata (CA), Agent-Based (AB) and a hybrid CA-AB (e.g., Mustafa et al., 2018, 2017; Vermeiren et al., 2016).

>Mustafa, A., Cools, M., Saadi, I., Teller, J., 2017. Coupling agent-based, cellular automata and logistic regression into a hybrid urban expansion model (HUEM). Land Use Policy 69, 529–540.

>Mustafa, A., Heppenstall, A., Omrani, H., Saadi, I., Cools, M., Teller, J., 2018. Modelling built-up expansion and densification with multinomial logistic regression, cellular automata and genetic algorithm. Computers, Environment and Urban Systems 67, 147–156.

>Vermeiren, K., Vanmaercke, M., Beckers, J., Van Rompaey, A., 2016. ASSURE: a model for the simulation of urban expansion and intra-urban social segregation. International Journal of Geographical Information Science 30, 2377–2400.

-Line 143: "The DEM data were extracted from the SRTM 90m resolution and, after resampling, all data were processed into 1×1 km2 raster files." Why do you resample the data to such low resolution? and what is the resample method do you employ?

-Line 146: "All distance measures were extracted from LUCC datasets from the years 2000 and 2015 using ArcGIS Euclidean distance". Euclidean distance is a basic GIS process that can be performed by many tools. No need to mention specific software for such a basic GIS analysis.

-Table 1 presents data with inconsistent dates (2000, 2015, or 2000, 2010). Please justify as this will bias the results.

-Line 207: "In our case study, 18,190 pixels (about 10% of the total) were selected by different sampling methods (Fig. S 3) to train (66% of the sample size) and test (34% of the sample size) the model." Please provide more details about your sample. Is it a

binary (0 no changes, 1 changes) excluding grassland with no change between 1975 and 2015?

-Figures 3 and 4: this evaluation of model performance was done for which period 1975-2000 or 2000-2015? AND do you consider all cells in the study are or the observed changes between two dates? Also, there is a sharp difference in performance between the Logit model and XGB, why? According to many studies that compared Logit with machine learning (ML) methods, ML outperformed logit but not such huge differences as presented in this study.

-Figure 6: can you present the variables' importance (Odds ratio) of the logit model as well? This will help readers to understand the differences between the two methods.

-Figure 9: I am confused about this probability map. I see that almost all pixels have a probability of either 100% (1) or 0% (0). So, is it really a gradient probability map? Another fundamental question, if we need to simulate future scenarios that assume a change of 100 pixels out of 1000 pixels (as an example) then this map is not useful as many pixels have a probability value of 100%. Should the model make a random selection from pixels with a 100% probability??

-English needs improvements.

---

## Short Comment (SC2) · 22 Jul 2020

Dear Dr. Astrid Kerkweg
Thank you for your comments on python script DOI and the datasets in our paper. According to your comments, I have published the python code at GitHub and Zenodo. The data also has been described clearly in GitHub README.md. The results in this paper could be reproduced by using data in GitHub.
Please check the following link of the python script:
Link: https://zenodo.org/record/3937226.Xw2M6egzZPY
DOI: 10.5281/zenodo.3937226
I have given the specification in the manuscript, please check the attached file.
Please check line 284-285, 560-563.

[Figure]

Thanks again for your professional comments!
Any question please contact me.

Kind regards
Batu

––––––––––––––––––––––––––––––

---

## Referee Comment (RC2) · Anonymous Referee #1 · 6 Aug 2020

I think the available manuscript on the platform is not the updated one.

---

## Short Comment (SC3) · 6 Aug 2020

**Response to comments**

This manuscript tests whether XGBoost can provide alternative insights that conventional land-use models are unable to generate. The overall methodology is interesting. I have a number of major comments before I can suggest the paper for publication.

We appreciate the thoroughness with which you went through our manuscript. We consider all your comments as very useful, even though we may have disagreed here and there. We believe that with the help of your work, this manuscript has further improved. Thank you!

1, Line 54: "Some such models are spatial (e.g. CLUE-S, GeoSOS-FLUS, LTM, Fu et al., 2018; Liang et 55 al., 2018; Pijanowski et al., 2002, 2005; Verburg & Veldkamp, 2004; Zhang et al., 2013); others are not (e.g. Markov models; Iacono et al., 2015; Yuan et al., 2015)." Authors should be aware that all land use change models are spatial models. Markov models are used to estimate the quantity of change from one land use state to another but are not land use change simulators.

**Response:** Thank you for your interesting discussion. In fact, we were to say that some models are spatially explicit, and some are not. We have corrected this in line 55-56. Markov models are not spatially explicit and can deal with numbers without any spatial relation. However, we agree with you that of course all land-use models refer to a spatial concept.

2, Line 57: "Hybrid models, which combine different approaches to make the best use of the advantages of each model, are another important variety. This type of model is used to characterize the multiple aspects of LUCC patterns and processes (Li and Yeh, 2002; Sun and Müller, 2013)." Authors did not discuss important other land use modeling approaches such as Cellular Automata (CA), Agent-Based (AB) and a hybrid CA-AB (e.g., Mustafa et al., 2018, 2017; Vermeiren et al., 2016).

Response: We have added a discussion about these models as you suggested in line 56-66.

3, Line 143: "The DEM data were extracted from the SRTM 90m resolution and, after resampling, all data were processed into  $1 \times 1$  km2 raster files." Why do you resample the data to such low resolution? and what is the resample method do you employ?

Response: A total of twenty drivers was used in this paper. Two of them, population density and GDP density, were at  $1 \times 1 \text{ km}^2$ , two other, temperature and precipitation, were at even coarser resolution (station-based).  $1 \times 1 \text{ km}^2$  seemed a good compromise between the finest and the coarsest resolution. In addition, the area of study region is  $20.4 \times 10^4 \text{ km}^2$ , for which a finer resolution would not have seemed appropriate. We resampled by using the NEAREST method in ArcGIS. We have added this in the manuscript at line 150.

4, Line 146: "All distance measures were extracted from LUCC datasets from the years 2000 and 2015 using ArcGIS Euclidean distance". Euclidean distance is a basic GIS process that can be performed by many tools. No need to mention specific software for such a basic GIS analysis. - Table 1 presents data with inconsistent dates (2000, 2015, or 2000, 2010). Please justify as this will

bias the results.

Response: With respect to reproducibility, we remain with giving the tool for computing the Euclidian distance. High-resolution population and GDP density were only available for 2000 and 2010. We have added a discussion on the bias that could be caused by this data. Please check line 160-164.

5, Line 207: "In our case study, 18,190 pixels (about 10% of the total) were selected by different sampling methods (Fig. S 3) to train (66% of the sample size) and test (34% of the sample size) the model." Please provide more details about your sample. Is it a binary (0 no changes, 1 changes) excluding grassland with no change between 1975 and 2015?

Response: As you suggested, we have added a more detailed description of the sample in line 216-221.

6, Figures 3 and 4: this evaluation of model performance was done for which period 1975-2000 or 2000-2015? AND do you consider all cells in the study are or the observed changes between two dates? Also, there is a sharp difference in performance between the Logit model and XGB, why? According to many studies that compared Logit with machine learning (ML) methods, ML outperformed logit but not such huge differences as presented in this study.

Response: Thank you for your detailed question.

1) The model performance was done for the newly added grassland in both periods

2) We are not surprised by this large difference. The tree-based models are always expected to outperform linear models. We have used 33% of the data for validation, which were not included in the training. So, over-training should not be an issue, so we have to assume that the difference between the linear and non-linear approach is responsible for this difference in performance.

**7, Figure 6: can you present the variables' importance (Odds ratio) of the logit model as well? This will help readers to understand the differences between the two methods.**

Response: As you mentioned, SHAP values as a statistical method could be combined with many other ML models to present the variables' importance. However, the Logistic regression model is not a robust model in simulating grassland degradation in this study. The kappa index is 0.72. To present variables' importance using such a weak model does not make any sense to us. The Logistic regression was used as benchmark in this study and has proven that a non-linear machine learning model could achieve a better predictive quality than linear methods. This is the aim of this study. We have put it down here for you, but we think that it is not providing any additional information for the reader. For this reason, we refrain from adding it to the manuscript.

Figure 1: Decomposed SHAP values for the individual prediction of an example pixel (Logistic regression model).

Figure 2: Driver ranking by SHAP values based on the training dataset (66% of sample size) using the over-sampling method (Logistic regression model).

---

## Referee Comment (RC3) · Anonymous Referee #2 · 28 Sep 2020

Dear authors

Thank you for your interesting submission. This paper presents an interesting suite of tools for investigating an important topic of research. I have only some minor comments to make before publication:

As with any ML interpretation, one wonders how more generally useful this is to other regions. Have you considered whether testing against historical datasets is worthwhile? In other words, have you tried to apply this method to re-analyse previously studied grassland degradation? If not, are there similar areas of focus this might work on?

Please can you comment on the compute hardware required for training?

Please include a Zenodo, or other archive, snapshot of the data used in this study.

Thanks!

---

## Author Comment (AC2) · 26 Oct 2020

Dear referee
We will share the revised manuscript after the editor close the discussion, now the discussion have closed,
We will share the manuscript soon.
Thank you so much for your comments.
Kind regards
Author
* * *
[Figure]

2020.

---

## Author Response (AR1)

**Author's response**

**Response to gmd-2020-59-RC1**

Dear Dr. Astrid Kerkweg

Thank you for your comments on python script DOI and the datasets in our paper.

According to your comments, I have published the python code at GitHub and Zenodo.

The data also has been described clearly in GitHub README.md. The results in this paper could be reproduced by using data in GitHub.

Please check the following link of the python script:

Link: https://zenodo.org/record/3937226#.Xw2M6egzZPY

DOI:   10.5281/zenodo.3937226

I have given the specification in the manuscript, please check the attached file.

Please check line 284-285, 560-563

Thanks again for your professional comments!

Any question please contact me.

Kind regards

Batu

**Response to gmd-2020-59-RC2**

This manuscript tests whether XGBoost can provide alternative insights that conventional land-use models are unable to generate. The overall methodology is interesting. I have a number of major comments before I can suggest the paper for publication.

We appreciate the thoroughness with which you went through our manuscript. We consider all your comments as very useful, even though we may have disagreed here and there. We believe that with the help of your work, this manuscript has further improved. Thank you!

1, Line 54: "Some such models are spatial (e.g. CLUE-S, GeoSOS-FLUS, LTM, Fu et al., 2018; Liang et 55 al., 2018; Pijanowski et al., 2002, 2005; Verburg & Veldkamp, 2004; Zhang et al., 2013); others are not (e.g. Markov models; Iacono et al., 2015; Yuan et al., 2015)." Authors should be aware that all land use change models are spatial models. Markov models are used to estimate the quantity of change from one land use state to another but are not land use change simulators.

**Response:** Thank you for your interesting discussion. In fact, we were to say that some models are spatially explicit, and some are not. We have corrected this in line 55-56. Markov models are not spatially explicit and can deal with numbers without any spatial relation. However, we agree with you that of course all land-use models refer to a spatial concept.

2, Line 57: "Hybrid models, which combine different approaches to make the best use of the advantages of each model, are another important variety. This type of model is used to characterize the multiple aspects of LUCC patterns and processes (Li and Yeh, 2002; Sun and Müller, 2013)." Authors did not discuss important other land use modeling approaches such as Cellular Automata (CA), Agent-Based (AB) and a hybrid CA-AB (e.g., Mustafa et al., 2018, 2017; Vermeiren et al., 2016).

Response: We have added a discussion about these models as you suggested in line 56-66.

3, Line 143: "The DEM data were extracted from the SRTM 90m resolution and, after resampling, all data were processed into 1×1 km2 raster files." Why do you resample the data to such low resolution? and what is the resample method do you employ?

Response: A total of twenty drivers was used in this paper. Two of them, population density and GDP density, were at $1\times1$ km$^2$, two other, temperature and precipitation, were at even coarser resolution (station-based). $1\times1$ km$^2$ seemed a good compromise between the finest and the coarsest resolution. In addition, the area of study region is $20.4\times10^4$ km$^2$, for which a finer resolution would not have seemed appropriate. We resampled by using the NEAREST method in ArcGIS. We have added this in the manuscript at line 150.

4, Line 146: "All distance measures were extracted from LUCC datasets from the years 2000 and 2015 using ArcGIS Euclidean distance". Euclidean distance is a basic GIS process that can be performed by many tools. No need to mention specific software for such a basic GIS analysis. -Table 1 presents data with inconsistent dates (2000, 2015, or 2000, 2010). Please justify as this will bias the results.

Response: With respect to reproducibility, we remain with giving the tool for computing the Euclidian distance. High-resolution population and GDP density were only available for 2000 and 2010. We have added a discussion on the bias that could be caused by this data. Please check line 160-164.

5, Line 207: "In our case study, 18,190 pixels (about 10% of the total) were selected by different sampling methods (Fig. S 3) to train (66% of the sample size) and test (34% of the sample size) the model." Please provide more details about your sample. Is it a binary (0 no changes, 1 changes) excluding grassland with no change between 1975 and 2015?

Response: As you suggested, we have added a more detailed description of the sample in line 216-221.

6, Figures 3 and 4: this evaluation of model performance was done for which period 1975-2000 or 2000-2015? AND do you consider all cells in the study are or the observed changes between two dates? Also, there is a sharp difference in performance between the Logit model and XGB, why? According to many studies that compared Logit with machine learning (ML) methods, ML outperformed logit but not such huge differences as presented in this study.

Response: Thank you for your detailed question.

1) The model performance was done for the newly added grassland in both periods

2) We are not surprised by this large difference. The tree-based models are always expected to outperform linear models. We have used 33% of the data for validation, which were not included in the training. So, over-training should not be an issue, so we have to assume that the difference between the linear and non-linear approach is responsible for this difference in performance.

7, Figure 6: can you present the variables' importance (Odds ratio) of the logit model as well? This will help readers to understand the differences between the two methods.

Response: As you mentioned, SHAP values as a statistical method could be combined with many other ML models to present the variables' importance. However, the Logistic regression model is not a robust model in simulating grassland degradation in this study. The kappa index is 0.72. To present variables' importance using such a weak model does not make any sense to us. The Logistic regression was used as benchmark in this study and has proven that a non-linear machine learning model could achieve a better predictive quality than linear methods. This is the aim of this study. We have put it down here for you, but we think that it is not providing any additional information for the reader. For this reason, we refrain from adding it to the manuscript.

[Figure]

Figure 1: Decomposed SHAP values for the individual prediction of an example pixel (Logistic regression model).

[Figure]

Figure 2: Driver ranking by SHAP values based on the training dataset (66% of sample size) using the over-sampling method (Logistic regression model).

[Figure]

[Figure]

Figure 3: The SHAP dependence plot for each driver (Logistic regression model).

8, Figure 9: I am confused about this probability map. I see that almost all pixels have a probability of either 100% (1) or 0% (0). So, is it really a gradient probability map?

It is, but in fact, the number of pixels that have values between 0 and 100 is small. On top, we have 86% of the pixels defined as grassland, which is why the map looks almost complete, but it is not. We have adjusted the map and included a zoom in to one region where more gradient values are located, please check in line 428-430

Another fundamental question, if we need to simulate future scenarios that assume a change of 100 pixels out of 1000 pixels (as an example) then this map is not useful as many pixels have a probability value of 100%. Should the model make a random selection from pixels with a 100% probability??

You are right, for prediction purposes this map is not useful. We have already discussed the difficulties that occur if you used such approach to simulate future or other hypothetical scenarios and concluded that the ML approach must be combined with other modelling approaches in order to be able to produce scenarios. In this case, we just demonstrate with this map the vulnerability of the region to further grassland degradation. The probability for grassland degradation is the closest we can get to a spatial explicit prediction using XGBoost and SHAP.

9, English needs improvements.

This manuscript has been reviewed by a professional British language editor for scientific publications.

**Response to gmd-2020-59-RC3**

1, Thank you for your interesting submission. This paper presents an interesting suite of tools for investigating an important topic of research. I have only some minor comments to make before publication:

Response: We appreciate your suggestions for our manuscript and we consider all your comments as very useful. We have addressed each of your comments below.

2, As with any ML interpretation, one wonders how more generally useful this is to other regions. Have you considered whether testing against historical datasets is worthwhile?

In other words, have you tried to apply this method to re-analyses previously studied grassland degradation? If not, are there similar areas of focus this might work on?

Response: Thank you for your useful question.

We have not yet been able to test this method in another region or to historical datasets.

But the method in this study has certainly the potential for transferability for two reasons:

First of all, we have used this method on another topic and dataset in the same region, namely for studying land degradation. Please see Land-use change and land degradation on the Mongolian Plateau from 1975 to 2015 — a case study from Xilingol, China. Land Degradation & Development 29: 1595–1606. DOI: 10.1002/ldr.2948. XGBoost and SHAP presented an excellent performance as well.

Secondly, the datasets in this study (land use and driver data) are public and available and could be replaced by other datasets. For more information please see the data description in the manuscript, line 171-173.

Based on this, we believe XGBoost and SHAP provide large potential to be applied to other datasets, regions and topics as well.

Actually, the datasets in this manuscript are historical data and we predict the dynamic grassland degradation (newly added grassland degradation, NGD) between 2000-2015 in Xilingol based on historical data from 1975 to 2015. Then we use the historical data from 2000-2015 to test the predicted NGD between 2000-2015. The results indicated that it is worthwhile testing against the historical data.

More detailed information about the data in this manuscript please check line 143-170. In addition, for a ML model,

3, Please can you comment on the computer hardware required for training?
Response: Done.

I comment the computer hardware in "Code and data availability" section, please check
line 582-594.

The used XGBoost algorithm including the SHAP library runs well on a modern (Intel
or AMD) PC (4 cores or more, 16 GB RAM). The training and the simulation were
made on Linux as operating system but should work also under Windows.

4, Please include a Zenodo, or other archive, snapshot of the data used in this study.
Thanks!

Response: Thank you for your careful comments, I have published the python code at
GitHub and Zenodo. The data also has been described clearly in GitHub README.md.
The results in this paper could be reproduced by using data in GitHub.

Please check the following link of the python script:

Link: https://zenodo.org/record/3937226#.Xw2M6egzZPY

DOI:   10.5281/zenodo.3937226

I have given the specification in the manuscript, please check the attached file.

Please check line 284-285, 560-563

[revised manuscript text omitted]